# Fatty Acid Profile of Erythrocyte Membranes in Patients with Psoriasis

**DOI:** 10.3390/nu16121799

**Published:** 2024-06-07

**Authors:** Mariola Marchlewicz, Zofia Polakowska, Dominika Maciejewska-Markiewicz, Ewa Stachowska, Natalia Jakubiak, Magdalena Kiedrowicz, Aleksandra Rak-Załuska, Michał Duchnik, Alicja Wajs-Syrenicz, Ewa Duchnik

**Affiliations:** 1Department of Dermatology and Venereology, Faculty of Health Sciences, Pomeranian Medical University, 71-210 Szczecin, Poland; 2Department of Aesthetic Dermatology, Faculty of Health Sciences, Pomeranian Medical University, 70-111 Szczecin, Poland; 3Department of Human Nutrition and Metabolomics, Pomeranian Medical University, 71-460 Szczecin, Poland; 4Department of General and Vascular Surgery, Public Voivodeship Combined Hospital, 70-891 Szczecin, Poland

**Keywords:** psoriasis, omega-3, EPA, DHA, PUFA, SFA

## Abstract

Psoriasis is a chronic systemic disease with a multifaceted pathomechanism and immunological basis, with the presence of inflammatory skin lesions and joint ailments. Diseases accompanying psoriasis include metabolic and cardiovascular disorders. It has been suggested that inflammation is involved in the development of each of these conditions. The main objective of this study was to analyse the fatty acid profile, including polyunsaturated fatty acids, in the erythrocyte membranes of patients suffering from psoriasis. A total of 58 adult patients of the Department of Skin and Venereal Diseases of the Pomeranian Medical University in Szczecin, suffering from psoriasis, were qualified for this study. The patients had undergone an interview and physical examination, during which the severity of psoriasis was assessed. All patients had their weight and height measured to assess their body mass index (BMI). After 3 months of treatment, biochemical parameters (ALT, AST, total cholesterol) and inflammatory markers (CRP) in the blood were assessed. In addition, the isolation of fatty acids (PUFAs, SFAs, MUFAs) from erythrocyte membranes and the qualitative and quantitative analysis of their profile using a gas chromatograph were carried out. In patients with severe psoriasis requiring systemic treatment, an altered profile of fatty acids in erythrocyte membranes was found, including a significantly lower concentration of polyunsaturated fatty acids (omega-3), which have an anti-inflammatory effect; a significantly higher concentration of saturated fatty acids; and a decreased concentration of oleic acid (omega-9), compared to the results obtained in patients with less severe psoriasis receiving topical treatment. In patients with psoriasis and BMI ≥ 25, significantly higher concentrations of AST and ALT in the blood and significantly higher concentrations of pro-inflammatory arachidonic acid in erythrocyte membranes were found. Elevated concentrations of saturated (R = 0.31) and monounsaturated fatty acids (R = 0.29) may correlate with a greater severity of psoriasis.

## 1. Introduction

Psoriasis is a systemic disease with a complex pathomechanism, manifesting with inflammatory skin lesions and joint complaints. Comorbidities include metabolic disorders, such as obesity and diabetes, and cardiovascular diseases, e.g., hypertension and/or coronary heart disease. Although the systemic nature of the disease is not always recognised, it has been suggested that the development of psoriasis comorbidities, which significantly affect patients’ health and quality of life, may be due to inflammation [1]. 

There are two commonly acknowledged phases for the pathogenesis of psoriasis: the initiation/triggering of the disease, and the maintenance of the pathological status. The initiation phase involves genetic predisposition and environmental factors that trigger the onset of psoriasis, such as infections, skin trauma, or stress. The maintenance phase is characterised by a chronic inflammatory response, mediated by immune cells like T-cells, dendritic cells, and keratinocytes, which sustain the disease through a complex network of cytokines and signalling pathways [2]. The pathomechanism of the disease involves pro-inflammatory mediators, which are also responsible for excessive keratinocyte proliferation. These include TNF-α, IL-17, IL-22, IL-23, and IFN-α, with the IL-17/IL-23 axis appearing to play a key role. Persistent inflammation is a consequence of impaired activity of both the innate and adaptive immune responses [3]. 

Omega-3 fatty acids have a strong anti-inflammatory effect, inhibiting IL-1β, TNF-α, and IL-6, while omega-6 fatty acids are mainly pro-inflammatory. It has been suggested that an appropriate ratio of omega-3 to omega-6 fatty acids, and consequent changes in the ratio of anti-inflammatory to pro-inflammatory eicosanoids, may modulate the symptoms of inflammatory diseases, including psoriasis. It therefore appears that the type and amount of omega-3 and omega-6 fatty acids supplied to the body may play an important role in the development and progression of this condition [4]. 

The aim of this study was to analyse the fatty acid profile, including polyunsaturated fatty acids, in erythrocyte membranes of psoriasis patients, in relation to body mass index and disease severity. 

## 2. Material and Methods 

### 2.1. Study Design

This study aimed to include 100 patients (no power test was used to calculate sample size); unfortunately, most of the patients admitted to the clinic met the exclusion criteria. The study group consisted of 58 adult psoriasis patients under the care of the Department of Skin and Venereal Diseases at the Pomeranian Medical University in Szczecin. Participation in this study was open to adults of both sexes. Inclusion criteria: diagnosed psoriasis, age 18–65, and all types of psoriasis treatment. Exclusion criteria included factors and conditions that may affect the fatty acid profile in erythrocyte membranes, including smoking; alcohol and drug abuse; coronary heart disease; type 2 diabetes and other systemic diseases; and the use of hypolipidemic agents, dietary supplements, and antioxidants. This study was approved by the Bioethics Committee at the Pomeranian Medical University. Approval number KB-006/22/2022. Eligible patients provided their medical history, underwent a physical examination, and were assessed for psoriasis severity using the following criteria: Psoriasis Area and Severity Index (PASI), body surface area (BSA), Dermatology Life Quality Index (DLQI). All patients had their weight and height measured to assess body mass index (BMI). Subsequently, blood samples were taken from them for laboratory tests including CRP (C-reactive protein), liver enzymes, and total cholesterol. 

### 2.2. Isolation of Fatty Acids from Erythrocytes 

For fatty acid analysis, 5 mL of blood was collected from all patients and placed into heparin tubes. The blood was then centrifuged (1850× *g*, 4 °C, 10 min) to separate erythrocytes from plasma. An amount of 0.5 mL of cells was transferred to a new tube, adding 10 mL of buffered 0.9% sodium chloride solution, pH 7.4 (PBS). The contents of the tube were washed three times to obtain a clear supernatant. Then, 13 mL of bidistilled water and 200 µL of butylated hydroxytoluene (BHT) were added. For erythrocyte haemolysis, the sample was incubated for 5 min at −80 °C and then centrifuged for 60 min (1850× *g*, 4 °C). In the final step, the supernatant was removed [5]. 

The extraction of erythrocyte membranes was performed according to the modified Folch method [6]. The lipid layer was obtained by adding 3 mL of Folch’s solution, containing chloroform and methanol in a 2:1 ratio. Internal standard (C21:0, 2 mg/mL) and BHT were then added at 100 µL each and incubated for 40 min on a shaker at room temperature. Finally, the solution was centrifuged to remove any impurities. The lipid layer was transferred to a new tube. Fatty acid esters were saponified with 1 mL of 2 M KOH in methanol, followed by incubation for 20 min at 20 °C, the addition of 2 mL of a 14% solution of boron trifluoride in methanol, and re-incubation, maintaining the previous conditions. The final step was to isolate the resulting methylene esters using 2 mL of hexane and 10 mL of sodium chloride. This resulted in a separation between the aqueous layer (bottom) and the organic layer (top), and 1 mL of hexane (organic layer) was collected in a glass vial. 

### 2.3. Fatty Acid Analysis by Gas Chromatography with Flame Ionisation Detector (FID) 

Fatty acid analysis was performed with a gas chromatograph (Agilent Technologies 7890A GC System) using a 15 m × 0.10 mm, 0.10 μm capillary column (Supelcowax™ 10 Capillary GC Column, Supelco, Bellefonte, PA, USA). The following conditions were used for chromatographic separation: The initial temperature was 60 °C, followed by a 40 °C/min increase to 160 °C and a further 30 °C/min increase to 190 °C. After that, the increase continued at 30 °C/min until 230 °C was reached. The final temperature was 230 °C. The flow rate of the carrier gas (hydrogen) through the column was 0.8 mL/min. 

For qualitative identification of fatty acids, retention times were compared with standards. ChemStation Software B.04.03 (Agilent Technologies, Cheshire, UK) was used for quantitative analysis. The percentage concentrations of the individual fatty acids were calculated against the internal standard, which was heneicosanoic acid (C21:0). 

### 2.4. Statistical Analysis

Statistical analysis was performed using the R software package, version 3.0.2. The Shapiro–Wilk test was used to check whether the data distribution was normal. As the data distribution was not different from normal, parametric tests were used. To test the differences between groups for statistical significance, Student’s *t*-test for unpaired data was used. Correlation analysis between individual parameters and fatty acids was performed using Pearson’s correlation test. The statistical significance level was set at *p* < 0.05.

## 3. Results

### 3.1. Characteristics of the Respondents

The study group consisted of 58 adult patients, 64% of whom were male (*n* = 37) and 36% were female (*n* = 21). Patients had a mean PASI score of 8.28 and mean DLQI score of 8.98. The average percentage of body surface area affected by psoriatic lesions was 15.94%. Depending on the severity of the disease and contraindications, the study group received topical or systemic treatment (cyclosporine A at 3–5 mg/kg b.w./day or methotrexate at 15 mg/week). Respectively, 51% of the patients (*n* = 30) received topical treatment (PASI < 10) and 49% (*n* = 28) received systemic treatment for three months (PASI ≥ 10). 

The mean BMI of the study participants was 27.26 kg/m^2^. Patients in the study group had elevated levels of C-reactive protein (CRP), with a mean level of 17.24 mg/L ± 29.06. All other laboratory results were normal. The test results are given in detail in Table 1.

The study group underwent both qualitative and quantitative analysis of the fatty acid profile. The most abundant fatty acids in erythrocyte membranes were found to be palmitoleic acid, with a mean content of 35.93% ± 1.80, and stearic acid, with a mean content of 30.90% ± 3.12, while the least abundant were EPA, with a mean content of 0.49% ± 0.28; ALA, with a mean content of 0.63% ± 0.31; and DHA, with a mean content of 0.92% ± 0.64. The average concentrations of all fatty acids are shown in Table 2. 

### 3.2. Comparison of the Fatty Acid Profile and Selected Clinical Parameters in Relation to BMI 

The study group was divided into two subgroups based on BMI. The first subgroup, which accounted for 45% (*n* = 26) of the study group, was made up of patients with a normal body mass index (BMI < 25). The other subgroup included patients with an elevated BMI (≥25), who accounted for 55% (*n* = 32) of the study group.

Patients with an elevated BMI were observed to have statistically significantly higher levels of alanine aminotransferase (ALT), with a mean value of 15.38 U/L ± 5.36 in patients with BMI < 25 vs. 40.00 U/L ± 22.22 in patients with BMI ≥ 25 (*p* = 0.0007); and aspartate aminotransferase (AST), with a mean value of 20.13 U/L ± 6.75 in patients with BMI < 25 vs. 32.69 U/L ± 14.90 in patients with BMI ≥ 25 (*p =* 0.012). No significant differences were observed in the other parameters. The results are presented in detail in Table 3. 

Differences in the fatty acid content of erythrocyte membranes between the groups were also analysed. An increased BMI was significantly associated with an increased mean percentage content of arachidonic acid, 4.08 ± 0.99 vs. 3.43 ± 0.99 (*p* = 0.02), in patients with normal BMI. No significant differences were observed for the other acids, as shown in Table 4.

The researchers also analysed the differences in the levels of saturated, monounsaturated, and polyunsaturated fatty acids between the groups of patients with normal and increased BMI. However, no statistically significant differences were observed in the Student’s *t*-test. The data are presented in Table 5.

### 3.3. Comparison of the Fatty Acid Profile and Selected Clinical Parameters in Relation to the Treatment Used 

Significantly lower BSA scores were observed in the subgroup of patients who received 3 months of systemic treatment (10.25% ± 7.13) compared to the patients treated topically (22.92% ± 25.68) (*p* = 0.02). The systemic treatment subgroup had a significantly increased body mass index compared to the subgroup on topical treatment only, with a mean of 30.33 kg/m^2^ ± 7.54 vs. 24.4 kg/m^2^ ± 7.77 (*p* = 0.01). The results are shown in detail in Table 6. 

Significant differences were found between the subgroups in the percentage of oleic acid content. Compared to the systemic treatment subgroup, patients receiving topical treatment had a higher percentage of oleic acid in erythrocyte membranes (mean 14.47% ± 2.44 vs. 13.36% ± 2.19, *p* = 0.04). In addition, a statistical trend was observed of reduced DHA (*p =* 0.07) and EPA (*p* = 0.06) in the subgroup of patients receiving systemic therapy, compared to the topical treatment subgroup. The results are presented in Table 7. 

Patients receiving systemic treatment had significantly higher mean concentrations of saturated fatty acids in erythrocyte membranes than the group receiving topical therapy (34.19% ± 3.14 vs. 31.45% ± 2.61, *p =* 0.01). There was a statistical trend for lower mean levels of polyunsaturated fatty acids in the systemic treatment subgroup (12.14% ± 1.91), compared to the subgroup receiving topical treatment (13.07% ± 1.89) (*p* = 0.07). No statistically significant differences were observed in the percentage of monounsaturated fatty acids, as shown in Table 8. 

### 3.4. Correlation Analysis of Selected Clinical and Biochemical Parameters and the Fatty Acid Profile

The final step of this study was to analyse the correlations between selected clinical parameters and the fatty acid profile (Table 9). 

A positive correlation was observed between stearic acid (*p =* 0.02, FDR = 0.05), palmitic acid (*p =* 0.02), and the saturated (*p =* 0.02, FDR = 0.04) and monounsaturated fatty acid groups (*p =* 0.05, FDR = 0.06) and the BSA index in psoriasis patients. 

Stearic acid content was also positively correlated with alanine aminotransferase (*p =* 0.02, FDR = 0.06). Furthermore, an increased saturated fatty acid content in erythrocyte ghosts was shown to be positively correlated with alanine aminotransferase (*p* = 0.05, FDR = 0.06).

The other clinical parameters were not correlated with the fatty acid profiles tested. 

## 4. Discussion

Current research into the fatty acid profile in psoriasis patients focuses mainly on determining their serum concentrations, and there is a lack of studies on FA levels in erythrocyte membranes. According to the literature, serum is a sensitive indicator of changes in lipid profile that have occurred over the course of several days, whereas erythrocyte membranes, due to their long lifespan, reliably reflect long-term changes in body fatty acid concentrations [7]. To date, the only publication on this topic is a study by Yaman et al., who conducted a pilot study in a group of 30 psoriasis patients. The aim was to determine the ratio of omega-6/omega-3 fatty acids in erythrocyte membranes and then to analyse their potential relationship with selected inflammatory mediators [4]. Statistically significant differences were found in the inflammatory parameters, including higher levels of CRP and IL-6, compared to the control group (n = 36). Furthermore, the ratio of omega-6/omega-3 fatty acids in erythrocyte membranes was significantly higher in the psoriatic group and correlated positively with inflammatory markers. The authors reported that patients with psoriasis had lower concentrations of *n*-3 fatty acids in erythrocyte membranes (by 12.5%) and higher concentrations of *n*-6 fatty acids (by 9.2%) compared to controls. Although no significant changes in docosahexaenoic acid (*n*-3) were observed, eicosapentaenoic acid (*n*-3) levels were significantly lower in patients with psoriasis. In a further part of this study, the authors divided the psoriatic group into two subgroups based on the median PASI score. Interestingly, the *n*-6/*n*-3 ratio and CRP levels were shown to be statistically significantly higher in the group with a median PASI score ≥ 3.85, whereas no significant differences were observed in the group with a median PASI score < 3.85, suggesting that a higher *n*-6/*n*-3 ratio may be associated with a more severe clinical course of psoriasis. 

Myśliwiec et al. [8] undertook a study to measure serum concentrations of fatty acids, including polyunsaturated fatty acids, and to determine their associations with inflammatory markers, disease severity, and the potential involvement in psoriatic comorbidities, e.g., type 2 diabetes, obesity, and hypertension. This study enrolled 85 patients, including 28 women and 57 men with exacerbated plaque psoriasis, and 32 healthy controls matched for age and gender. The authors observed significantly lower concentrations of PUFAs in the psoriatic group and a link between PASI score and reduced levels not only of DHA but of the entire group of *n*-3 PUFAs. Moreover, there was a positive correlation between PASI and the *n*-6/*n*-3 ratio in the entire psoriatic group. This study was the first to find a negative correlation between the content of docosahexaenoic acid and a more severe course of psoriasis in non-obese patients (BMI < 30). The authors suggested that low levels of *n*-3 PUFAs may be associated with an enhanced inflammatory response and, at the same time, a more severe clinical course of the disease [8]. In addition, a lower percentage of PUFAs, a higher percentage of SFAs, and a higher ratio of saturated to unsaturated fatty acids were found in psoriasis patients with comorbid hypertension. Similar results were found in another study by the same authors, with abnormal serum fatty acid profiles, including reduced levels of both *n*-3 and *n*-6 fatty acids, in patients with psoriasis and psoriatic arthritis (PsA) compared with controls [9]. Low levels of docosahexaenoic acid, eicosapentaenoic acid, and α-linolenic acid may have predisposed patients with PsA to a more severe clinical phenotype and higher levels of inflammatory markers. In turn, Qin et al. [10], in their study using transgenic mice as a model of human psoriatic lesions, observed an association between endogenous *n*-3 PUFAs and modulation of the inflammation underlying psoriasis, including a reduction in the population of Th cells and levels of IL-17, IL-22, and IL-23 [10].

The focus of the present study was to analyse the fatty acid profile in the erythrocyte membranes of psoriasis patients. Of all the fatty acids, the highest percentage content in erythrocyte membranes was observed for palmitoleic acid (35.93%), a monounsaturated fatty acid, and stearic acid (30.90%), a saturated fatty acid. On the other hand, the least abundant fatty acids in erythrocyte membranes were EPA (0.49%), ALA (0.63%), and DHA (0.92%), which belong to the omega-3 family and can inhibit the production of pro-inflammatory cytokines (TNF, IL-1β, IL-6). Therefore, their lower levels may correlate with more severe inflammation in psoriasis. The levels of EPA and DHA observed in psoriasis patients in this study are lower than those reported in the literature for healthy subjects, while the levels of stearic acid are higher [11]. 

The patients enrolled in this study had an average PASI of 8.28 points, DLQI of 8.98 points, and BSA of 15.94%. The mean level of CRP in the study group was elevated at 17.24 mg/L, clearly indicating an exacerbation of inflammation. This result is in line with the findings of Yamen et al. [12]. The average BMI of the patients enrolled in this study was 27.26 kg/m^2^, indicating overweight or obesity. Other biochemical parameters of the patients, such as liver enzymes and cholesterol levels, were normal. 

It should be noted that omega-3 fatty acids, particularly EPA and DHA, are also the main precursors of pro-resolving mediators responsible for quelling inflammation, such as protectins and D- and E-series resolvins. Their reduced levels seen in psoriasis patients (in the current study group) may compromise the process, leading to the resolution of the inflammatory response. Low levels of ALA also appear to be an unfavourable factor. However, research suggests that an improvement may be achieved by supplementation with omega-3 fatty acids. This is supported by recent meta-analyses documenting the association of docosahexaenoic acid and eicosapentaenoic acid with a reduction in C-reactive protein levels [13,14]. 

In turn, Morin et al. [15] in their study using a 3D model of psoriatic skin showed that docosahexaenoic acid supplementation had an inhibitory effect on the abnormal activity of psoriatic keratinocytes and reduced the secretion of TNF-α. Supplemented DHA was incorporated into the lipid fraction of the epidermis and used as a source of eicosapentaenoic acid. When added to the culture medium, it caused a reduction in pro-inflammatory eicosanoids derived from omega-6 acids, such as PGE2 and 12-HETE. A subsequent study by the same research team documented similar results for ALA in the same skin model [16]. The authors showed that the administration of exogenous α-linolenic acid inhibited the production of inflammatory cytokines by activated T cells, reduced keratinocyte hyperproliferation, and decreased levels of tumour necrosis factor alpha [16]. In conclusion, the reduced levels of docosahexaenoic acid, eicosapentaenoic acid, and α-linolenic acid found in the present study may explain the persistent inflammatory environment in patients with psoriasis, evidenced by an elevated CRP. However, it should be noted that the patient population in this study was on average overweight, which may be an additional factor promoting the development of psoriatic lesions. The relationship between obesity and psoriasis is described in more detail in the discussion below [14].

According to a prospective study by Setty et al. [17], weight gain and obesity are predisposing factors for psoriasis in women. On the other hand, the analysis by Armstrong et al. [18] demonstrated that people with psoriasis have a higher risk of developing obesity than the general population. In addition, patients with severe psoriasis have a higher risk of developing obesity than those with mild psoriasis. These relationships confirm the link between these conditions [12]. 

As mentioned above, there is a correlation between a reduced risk of developing cardiovascular and metabolic diseases, including obesity, and the consumption of a so-called Mediterranean diet, rich in *n*-3 and *n*-6 fatty acids [19,20,21]. According to the literature, these conditions are closely associated with psoriasis. In addition, research suggests that maintaining a healthy body weight and losing excess weight may reduce the risk of psoriasis and improve its clinical course [22,23]. 

Obesity and psoriasis are likely to interact at a functional level [18]. The potential mechanisms responsible for the exacerbation of skin lesions by obesity are likely to include interactions involving adipocytokines, adipokines, and fatty acids. The excessive size and number of adipocytes seen in obesity on the one hand stimulate pro-inflammatory adipokines, such as leptin and visfatin, and on the other, reduce anti-inflammatory adipokines, such as adiponectin [24]. Furthermore, Hafidi et al. [25] demonstrated that uncontrolled adipocyte hypertrophy is a source of free fatty acids, the levels of which are directly proportional to the degree of obesity [24,25]. 

In light of the above, the secondary aim of this study was to perform a comparative analysis of the fatty acid profile in erythrocyte membranes and selected clinical parameters in psoriasis patients in relation to body weight. The study group was divided into two subgroups: normal (BMI < 25) and overweight (BMI ≥ 25). Significantly higher mean liver enzyme levels were observed in patients with BMI ≥ 25, with a mean ALT of 40.00 U/L compared to 15.38 U/L in the BMI < 25 group (*p* = 0.0007) and a mean AST of 32.69 U/L compared to 20.13 U/L (*p* = 0.012). No significant differences were observed in the other biochemical parameters. 

Literature data confirm that obesity is directly associated with lipid disorders, impaired liver function, and hepatic steatosis. Free fatty acids released from visceral adipose tissue and dietary fatty acids directly contribute to liver damage, which may explain the higher mean liver enzyme levels observed in the group with higher BMI [24]. On the other hand, current research has shown that not only obesity, but also psoriasis is linked to lipid abnormalities. The study by Myśliwiec et al. [8] demonstrated that abnormal fatty acid profiles can occur in the course of psoriasis independently of obesity. According to the analysis by Kozłowska et al. [26], altered lipid profiles may predispose patients to liver dysfunction. On this basis, it appears that the overweight and obesity in patients with psoriasis may further predispose them to liver dysfunction. However, further long-term studies in larger populations are needed to fully elucidate the pathogenesis of this phenomenon. 

In the present study, a comparative analysis of the fatty acid profile revealed a statistically significant difference in the mean content of arachidonic acid, which was significantly higher in the group of patients with higher body weight (4.08%) compared to the group with a BMI < 25 (3.43%). To investigate the potential contribution of specific groups of fatty acids to obesity and overweight, the percentage content of saturated fatty acids, monounsaturated fatty acids, and polyunsaturated fatty acids was analysed in relation to BMI, but no significant differences were found. 

The present findings are consistent with reports that an elevated *n*-6/*n*-3 ratio promotes the development of obesity [27]. Guida et al. [22], in a study of obese patients with plaque-type psoriasis undergoing systemic treatment, documented that a low-energy diet enriched in *n*-3 fatty acids at an average dose of 2.6 g/d and with a reduced *n*-6 intake significantly improved clinical outcomes with lower PASI and itch scores, reductions in body weight and waist circumference, as well as lower serum triglycerides and total cholesterol. Omega-3 and omega-6 polyunsaturated fatty acids modify biochemical pathways, affecting lipid metabolism and organ function. The findings of Guida et al. [22] suggest that reducing the intake of omega-6 fatty acids in favour of omega-3 fatty acids may not only contribute to the direct relief of psoriatic lesions, but may also promote weight loss. 

The aim of psoriasis treatment is to halt the progression of the disease, reverse its symptoms, and improve the quality of life of patients. The choice of therapy is determined primarily by the severity of the disease and the individual requirements of the patient’s body, including contraindications. Topical treatment is used mainly for mild psoriasis as the main treatment or in combination with other treatment modalities. Systemic treatment is reserved for people with more severe forms of the disease.

It has been suggested that nutritional support in the form of omega-3 fatty acid supplementation may contribute to the more effective treatment of psoriasis. The literature suggests that people with psoriasis should not only avoid saturated fats (which also increase the risk of cardiometabolic disease), but also replace them with omega-3 fatty acids, which have anti-inflammatory effects [28,29]. 

In light of the above, the next stage of the present study was to analyse the fatty acid profile and selected clinical parameters in erythrocyte membranes in relation to the treatment used in psoriasis patients. The subgroup of patients who received systemic treatment for 3 months had a significantly lower BSA (10.25% vs. 22.92%, *p* = 0.02) and an increased BMI (30.33 kg/m^2^ vs. 24.4 kg/m^2^
*p* = 0.01), compared to the group receiving topical treatment only. A higher BSA score indicates a more severe course of psoriasis, reflecting the body surface area affected by the disease. In turn, the higher mean body mass index seems to confirm the previously described association between obesity and a more severe clinical phenotype of psoriasis. It is also important to mention the negative impact of obesity on treatment efficacy. 

In the present study, quantitative fatty acid profile analysis showed a higher mean content of oleic acid in patients receiving topical treatment compared to the subgroup receiving systemic treatment. In the systemic treatment subgroup, the researchers observed a significantly higher mean concentration of SFAs, a statistical trend towards lower DHA and EPA, and a trend towards lower mean PUFA content in erythrocyte membranes. Polyunsaturated fatty acids have a strong immunomodulatory effect, with EPA and DHA being primarily involved in attenuating inflammation [30]. Therefore, their reduced levels may be associated with an impaired anti-inflammatory mechanism, which may result in a more severe course of psoriasis. This phenomenon appears to explain the observed statistical trend towards lower levels of polyunsaturated fatty acids, including those of the omega-3 family, in people eligible for systemic treatment, i.e., with more severe psoriasis. 

Oleic acid is a monounsaturated fatty acid of the omega-9 family, the main source of which is olive oil, an essential part of the Mediterranean diet. It is an important factor in reducing the risk of cardiovascular disease [31]. Farag and Gad [32] in their recent paper on the anti-inflammatory effects of monounsaturated fatty acids, including oleic acid, highlighted their beneficial effects in alleviating skin inflammation and in wound healing. Current research also suggests that oleic acid may be a potential future alternative to topical agents in inflammatory skin conditions [33]. Ishak et al. [34] conducted a study to determine the effects of *n*-3, *n*-6, and *n*-9 fatty acid preparations, derived from linseed, evening primrose, and olive oil, respectively, on wound healing processes. The authors noted that each of the fatty acid families included in the analysis, when applied topically in the form of an emulsion, promoted skin regeneration processes compared to the control group, but in each subgroup, the process was achieved by different mechanisms [34]. Unfortunately, there are no studies on the oleic acid content in erythrocyte ghosts of patients with psoriasis. According to Finucane et al. [35], dietary monounsaturated fatty acids, including oleic acid, can inhibit the production of IL-1β, which is elevated in insulin resistance and obesity and which is generally known to be involved in the pathomechanism of psoriasis. Furthermore, it was observed that patients consuming a higher percentage of MUFAs in their diet presented both lower PASI scores and CRP levels [36]. The results of the present study indicate that mean oleic acid concentrations are significantly lower in those with more severe psoriasis, who receive systemic treatment. This may confirm its beneficial effects, as reported by Barrea et al. [36].

In contrast to polyunsaturated fatty acids, saturated fatty acids (SFAs) are markedly pro-inflammatory. The molecular mechanism underlying the inflammatory cascade activated by SFAs includes activation of the Toll-like receptor or the LPS receptor, which then induces activation of the nuclear factor NF-κβ and *cyclooxygenase 2 (COX-2*) [37]. *COX-2* is the enzyme responsible for the conversion of arachidonic acid to pro-inflammatory prostaglandins. Unsaturated fatty acids (UFAs) inhibit COX-2 expression, while SFAs induce it. Both obese and lean mice fed a diet rich in saturated fatty acids developed severe psoriasiform skin inflammation in the study by Herbert et al. [38]. The authors demonstrated that SFAs sensitise myeloid cells, priming them for an enhanced immune response mediated by Toll-like receptors, which consequently further stimulated keratinocyte proliferation. Additionally, excluding SFAs from the diet attenuated the inflammatory response [38]. These findings appear to explain the higher levels of SFAs observed in the present study in the subgroup with more severe psoriasis requiring systemic treatment. Reducing the intake of saturated fatty acids in favour of unsaturated fatty acids may support therapy and help achieve better clinical outcomes in psoriasis patients. 

Studies investigating the effect of fatty acids on the topical and systemic treatment of psoriasis have highlighted in particular the beneficial role of omega-3 PUFAs in this process. Patients in the aforementioned study by Guida et al.undergoing immunosuppressive treatment had significantly more favourable biochemical and clinical outcomes, including a reduction in PASI and itch scores, following a low-energy diet rich in *n*-3 fatty acids and poor in *n*-6 fatty acids [22]. The prevalence of obesity in the study group was a significant issue; similarly, patients receiving the same treatment in the present study also had elevated BMI. Millsop et al. in a meta-analysis on the role of dietary supplements in the management of psoriasis showed that fish oils rich in polyunsaturated fatty acids improve the efficacy of other therapies such as retinoids, vitamin D analogues, or UVB phototherapy [39]. Balbás and her team evaluated the effects of DHA and EPA supplementation in patients with plaque psoriasis receiving topical treatment [40]. In the experimental group, there was a significant improvement in PASI scores, a reduction in itch scores, and an overall improvement in quality of life as measured by DLQI. According to the meta-analysis by Clark et al. [41], which measured the efficacy of omega-3 fatty acids in the treatment of psoriasis, supplementation with these compounds is a potentially effective tool for improving the outcome of psoriasis patients. In conclusion, a well-balanced diet supplemented with omega-3 and omega-6 fatty acids can be a valuable adjunct to the treatment of psoriasis.

The subject literature highlights the existence of a link between a reduced risk of developing cardiovascular and metabolic diseases (closely linked to psoriasis) and following a Mediterranean diet, rich in omega-3 and omega-6 fatty acids, although it is important to note that a link has also been documented between the severity of psoriasis and adherence to the same Mediterranean diet [19,20,21,42]. The influence of fatty acids and fatty acid-derived lipid mediators on disease progression and possible comorbidities has not been fully elucidated to date. Therefore, the final stage of this study aimed to determine the relationship between selected clinical parameters and the FA profile.

A positive, low correlation was observed between BSA and stearic acid (and palmitic acid, as well as SFAs and MUFAs. Diet-derived SFAs are recognised as important activators of the TLR4-mediated inflammatory cascade, which promotes adipocyte inflammation and the development of insulin resistance [43]. According to the literature, patients with psoriasis have a higher risk of developing liver dysfunction, including nonalcoholic fatty liver disease [26,44]. In addition, these correlations are consistent with the previously observed, significantly higher levels of SFAs in the subgroup receiving systemic treatment designed for patients with more severe psoriasis. Azzini et al. [45] obtained similar results. Patients with psoriasis showed a significant increase in palmitic acid (*p* < 0.01) and in total SFA (*p* < 0.05). Moreover, authors revealed significant direct moderate correlations between SFA and erythrocyte sedimentation rate (r = 0.445), the duration of disease (r = 0.403), and morning stiffness (r = 0.434).

Although studies indicate that monounsaturated fatty acids have a beneficial effect on the body, due to their anti-inflammatory and antioxidant effects, the present study revealed a positive correlation between PASI scores and MUFA content in patients. According to the literature, high serum concentrations of MUFAs may be related to the overactivity of stearoyl-CoA desaturase-1 (SCD1), the enzyme responsible for converting SFAs to MUFAs [46]. The study by Mika et al. [46] found an association between increased serum concentrations of MUFAs in patients with chronic kidney disease and the risk of cardiovascular disease, though their concentrations were not related to the dietary intake. Unfortunately, there is a lack of research on this topic using erythrocyte membranes. On the other hand, a link between high concentrations of MUFAs and lower PASI scores was also observed by the research team led by Myśliwiec [8]. More research is needed to clearly define the effects of MUFAs and their role in the course of psoriasis. 

In the same study, Myśliwiec et al. [6] reported a correlation between PASI and *n*-3 PUFA concentrations, which is in line with the present results. Similarly, Baran et al. [27] documented a negative correlation of PASI with EPA and DHA. The findings reported in the present study are consistent with the above and confirm that low levels of *n*-3 PUFAs are associated with a more severe clinical course of the disease, probably due to ineffective suppression of the inflammatory response. In addition, an inverse association has been documented between omega-3 fatty acid content in erythrocyte membranes and levels of inflammatory biomarkers such as IL-6 and CRP in patients with cardiovascular disease [47]. Supplementation with *n*-3 PUFAs is considered a good strategy for both the prevention and treatment of psoriasis comorbidities. Additional supplementation with omega-3 fatty acids may improve treatment response and quality of life in psoriasis patients.

## 5. Conclusions

Psoriasis patients have an altered fatty acid profile in erythrocyte membranes, including lower concentrations of PUFAs. Our study revealed that high concentrations of SFA and reduced concentrations of oleic acid (*n*-9) may be connected with a more severe course of psoriasis. Moreover, concentrations of saturated and monounsaturated fatty acids may correlate with greater psoriasis severity and abnormal biochemical findings. Due to the small group size and low correlation levels, we cannot make clinically useful conclusions; however, the recommendations for consuming the right proportions of fatty acids should be added to the treatment regimen of patients with psoriasis. More research is needed to confirm these observations and plan an appropriate supplementation strategy.

## 6. Limitations

Although the results of this study lead to interesting conclusions, several limitations should be noted. First, the patient group is relatively small; this study assumed a representative group of 100 patients, but most of the patients admitted to the clinic met the exclusion criteria, usually related to comorbidities. Since this study was single-centre, the number of patients included in this study was reduced. Second, the correlations obtained have low statistical power, so they should be approached with caution.

## Figures and Tables

**Table 1 nutrients-16-01799-t001:** Laboratory and clinical parameters of the study group.

Parameter	Mean	SD
PASI [points]	8.28	8.24
BSA [%]	15.94	19.68
DLQI [points]	8.98	6.07
ALT [U/L]	30.43	20.77
AST [U/L]	27.04	13.66
CRP [mg/L]	17.24	29.06
CHOL [mg/dL]	192.77	40.06
BMI [kg/m^2^]	27.26	8.12

SD—standard deviation.

**Table 2 nutrients-16-01799-t002:** Fatty acid percentage content in erythrocyte membranes of psoriasis patients independent of other parameters.

FA Content [%]	Mean	SD
Myristic acid (C14:1)	2.75	0.43
Palmitic acid (C16:0)	1.90	0.78
Palmitoleic acid (C16:1)	35.93	1.80
Stearic acid (C18:0)	30.90	3.12
Oleic acid (C18:1n9)	14.01	2.43
Vaccenic acid (C18:1)	1.85	0.34
Linoleic acid (C18:2n6c)	6.92	1.62
Alpha-linolenic acid (C18:3n3)	0.63	0.31
Arachidonic acid (C20:4n6)	3.69	0.89
EPA (C20:5n3)	0.49	0.28
DHA (C22:6n3)	0.92	0.64

SD—standard deviation.

**Table 3 nutrients-16-01799-t003:** Differences in patients’ clinical parameters in relation to BMI.

Parameter	BMI < 25		BMI ≥ 25		*p*
	Mean	SD	Mean	SD	
PASI [points]	9.43	10.24	7.10	5.66	0.38
BSA [%]	17.32	22.46	12.62	12.65	0.42
DLQI [points]	9.42	6.12	8.90	6.01	0.77
ALT [U/L]	15.38	5.36	40.00	22.22	0.0007
AST [U/L]	20.13	6.75	32.69	14.90	0.012
CRP [mg/L]	4.30	2.07	9.28	13.85	0.77
CHOL [mg/dL]	191.34	56.23	198.70	20.83	0.75

SD—standard deviation. The *p*-value calculated using Student’s *t*-test.

**Table 4 nutrients-16-01799-t004:** Differences in the percentage content of fatty acids in relation to BMI.

FA Content [%]	BMI < 25		BMI ≥ 25		*p*
	Mean	SD	Mean	SD	
Myristic acid (C14:1)	2.72	0.40	2.77	0.45	0.65
Palmitic acid (C16:0)	2.00	0.54	1.83	0.90	0.43
Palmitoleic acid (C16:1)	35.47	1.76	36.25	1.75	0.15
Stearic acid (C18:0)	30.85	3.26	30.94	3.03	0.94
Oleic acid (C18:1n9)	13.57	2.44	14.30	2.37	0.32
Vaccenic acid (C18:1)	1.80	0.35	1.89	0.33	0.38
Linoleic acid (C18:2n6c)	7.37	1.68	6.62	1.49	0.13
Alpha-linolenic acid (C18:3n3)	0.66	0.28	0.61	0.33	0.63
Arachidonic acid (C20:4n6)	3.43	0.99	4.08	0.71	0.02
EPA (C20:5n3)	0.51	0.30	0.48	0.26	0.67
DHA (C22:6n3)	0.97	0.42	0.88	0.75	0.60

SD—standard deviation. The *p*-value calculated using Student’s *t*-test.

**Table 5 nutrients-16-01799-t005:** Differences in the percentage content of selected groups of fatty acids in relation to BMI.

Parameter [%]	BMI < 25		BMI ≥ 25		*p*
	Mean	SD	Mean	SD	
Saturated FAs (SFAs)	31.21	7.87	32.76	3.22	0.42
Monounsaturated FAs (MUFAs)	50.87	11.87	55.21	3.15	0.13
Polyunsaturated FAs (PUFAs)	12.92	3.63	12.02	1.55	0.11

SD—standard deviation. The *p*-value was calculated using Student’s *t*-test.

**Table 6 nutrients-16-01799-t006:** Differences in clinical parameters in patients in relation to the treatment used.

Parameter	Topical Treatment		Systemic Treatment		*p*
	Mean	SD	Mean	SD	
PASI [points]	7.48	5.06	9.78	10.57	0.33
BSA [%]	22.92	25.68	10.25	7.13	0.02
DLQI [points]	10.58	4.03	7.76	7.34	0.19
ALT [U/L]	25	12.56	30.58	21.69	0.98
AST [U/L]	31	19.3	26	12.64	0.31
CRP [mg/L]	3.6	3.21	12.24	25.38	0.11
CHOL [mg/dL]	165.49	40.35	194.48	40.11	0.15
BMI [kg/m^2^]	24.4	7.77	30.22	7.54	0.01

SD—standard deviation. The *p*-value calculated using Student’s *t*-test.

**Table 7 nutrients-16-01799-t007:** Differences in the percentage content of fatty acids in relation to the treatment used.

Parameter [%]	Topical Treatment		Systemic Treatment		*p*
	Mean	SD	Mean	SD	
Myristic acid (C14:1)	2.67	0.33	2.79	0.52	0.29
Palmitic acid (C16:0)	1.76	0.86	2.00	0.68	0.22
Palmitoleic acid (C16:1)	36.42	1.55	35.71	1.92	0.13
Stearic acid (C18:0)	29.69	2.58	32.19	2.94	0.33
Oleic acid (C18:1n9)	14.47	2.44	13.36	2.19	0.04
Vaccenic acid (C18:1)	1.92	0.36	1.81	0.32	0.37
Linoleic acid (C18:2n6c)	7.31	1.79	6.40	1.11	0.95
Alpha-linolenic acid (C18:3n3)	0.71	0.41	0.60	0.29	0.42
Arachidonic acid (C20:4n6)	3.48	0.82	3.85	0.90	0.25
EPA (C20:5n3)	0.54	0.25	0.45	0.29	0.06
DHA (C22:6n3)	1.02	0.79	0.83	0.47	0.07

SD—standard deviation. The *p*-value calculated using Student’s *t*-test.

**Table 8 nutrients-16-01799-t008:** Differences in the percentage content of selected groups of fatty acids in relation to the treatment used.

Parameter [%]	Topical Treatment		Systemic Treatment		*p*
	Mean	SD	Mean	SD	
Saturated FAs (SFAs)	31.45	2.61	34.19	3.14	0.01
Monounsaturated FAs (MUFAs)	55.48	2.94	53.67	2.58	0.79
Polyunsaturated FAs (PUFAs)	13.07	1.89	12.14	1.91	0.07

SD—standard deviation. The *p*-value calculated using Student’s *t*-test.

**Table 9 nutrients-16-01799-t009:** Correlations between selected clinical and biochemical parameters and the fatty acid profile.

Parameter vs. Parameter	BSA	PASI	ALT	CRP	CHOL
Stearic acid (C18:0)	R = 0.31 *p* = 0.02 #(FDR = 0.05)	NC	R = 0.32 *p* = 0.02 #(FDR = 0.06)	R = 0.36 *p* = 0.01 #(FDR = NC)	R = 0.29 *p* = 0.05 #(FDR = NC)
Palmitic acid (C16:0)	R = 0.33 *p* = 0.02 * #(FDR = 0.04)	NC	NC	NC	NC
SFAs	R = 0.31 *p* = 0.02 * #(FDR = 0.04)	NC	R = 0.29 *p* = 0.05 #(FDR = NC)	N, NC	R = 0.32 *p* = 0.02 #(FDR = NC)
MUFAs	R = 0.29 *p* = 0.05 * #(FDR = 0.06)	NC	NC	NC	NC
EPA	NC	R = −0.29 *p* = 0.05 #(FDR = NC)	NC	NC	NC
ALA	NC	R = −0.32 *p* = 0.02 #(FDR = NC)	R = −0.32 *p* = 0.03 #(FDR = NC)	NC	NC

The *p*-value calculated using Student’s *t*-test; R—Pearson’s linear correlation coefficient; * *p* < 0.05 # FDR—false discovery rate, significant after multiple comparisons: Bonferroni correction.

## Data Availability

Data are contained within the article.

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
