# Peer review of "Fatty Acid Profile of Erythrocyte Membranes in Patients with Psoriasis"

_nutrients, 2024, doi:10.3390/nu16121799_

Round 1
Reviewer 1 Report
Comments and Suggestions for Authors
The researchers measured levels of fatty acids in erythrocyte membranes of patients with psoriasis. Patients with more severe psoriasis were found to have lower levels of omega-3 and omega-9 poly-unsaturated acids, and higher level of saturated fatty acids. This article adds important information to the body of literature on fatty acids in psoriasis, which is usually studied through serum levels. of fatty acids. See a few comments in the attached pdf file.

Author Response
I would like to thank all the reviewers for their comments and time spent on the manuscript evaluation. Thank you for giving us the opportunity to revise and improve our work. The manuscript has been improved in accordance with the recommendations of most reviewers. However, we were unable to complete some of the suggestions. I have great hope that the improved version of our manuscript will meet the reviewer's suggestions.
All of the changes were written in red color. The manuscript was improved in abbreviations and limitations. We added the graphical abstract.
Reviewer 2 Report
Comments and Suggestions for Authors
The manuscript may be an useful contribution to the journal.
However, few changes may be needed to improve the value of the manuscript.
sample size: the authors should explain how the sample sizes were determined, if they were based on power calculations, and elaborate on that.
were the patients included on what criterium: were the patients consecutive?
Abbreviations should be explicited at their first appearance, either in the abstract or the manuscript body. Currently, the abstract lacks proper abbreviations explanations. (EPA, ALA and not only these.. please check within the entire article, there are many such places)
2.1 section in M&Methods: authors should explain why were those exclusion criteria selected (lines 69-71)
Introduction section should elaborate more on the mechanistic aspects of psoriasis; PMID: 30744173 could be consulted for more detailed aspects on the involvemebt of inflammation in psoriasis, in order to offer the reader a better framework, in like with the main idea of the Introduction section.
section 3.4 should not only report in the significance of the correlations, but also on the strength of the correlations in each pairwise correlation (Lines 204-210). Authors should avknowledge that Pearson’s r places most correlations within low correlations category.
The degree/strengths of correlations should also be included in the Abstract, for consistency.
- PMID: 7699656 also has tackled the same topic and problematic. The common aspects and differences to that study should also be included in the Discussion section as well.
lines 460-480 should report on the degree of correlation (strong, moderate, low), should the case be for each correlation, not only the significance.
the Conclusion may be rephrased, as correlation (let alone low correlations) do not imply causation. However, the conclusion depicts a causal statement, that is not clear conclusion of the study, as for that another stydy design should be employed (prospective study with diet intervention). This is not the case, therefore the conclusion should be rephrased.
a clearly deliniated Limitations paragraph should be introduced, to state the limitations of the study (small sample, sample sizes criteria, unicentric study, low correlations found, etc).
also, the authors should clearly state the way they have dealed with multiple comparisons and what adjustments that have employed (e.g. Bonferroni correction) as there are multiple analyais on same sample and without correction the risk of finding a significance where it is actually not a true significance in the data is quite high. All these suggestions are meant to simply improve the manuscript, the quality of the presentation and the data analysis, and I do hope the authors will take them into consideration within an significantly improved version for revision.
Comments on the Quality of English LanguageMinor spelling issues
Author Response
I would like to thank all the reviewers for their comments and time spent on the manuscript evaluation. Thank you for giving us the opportunity to revise and improve our work. The manuscript has been improved in accordance with the recommendations of most reviewers. However, we were unable to complete some of the suggestions. I have great hope that the improved version of our manuscript will meet the reviewers suggestions.
All of the changes were written in red color.
- sample size: the authors should explain how the sample sizes were determined, if they were based on power calculations, and elaborate on that.
RE: section 2.1 and limitation were improved.
2. were the patients included on what criterium: were the patients consecutive?
Re: Section 2.1 was improved.
3. Abbreviations should be explicited at their first appearance, either in the abstract or the manuscript body. Currently, the abstract lacks proper abbreviations explanations. (EPA, ALA and not only these.. please check within the entire article, there are many such places)
Re: Abbreviations was added to manuscript.
4. section in M&Methods: authors should explain why were those exclusion criteria selected (lines 69-71)
Re: Section 2.1 was improved.
5. Introduction section should elaborate more on the mechanistic aspects of psoriasis; PMID: 30744173 could be consulted for more detailed aspects on the involvemebt of inflammation in psoriasis, in order to offer the reader a better framework, in like with the main idea of the Introduction section.
Re: Introduction was improved.
6. section 3.4 should not only report in the significance of the correlations, but also on the strength of the correlations in each pairwise correlation (Lines 204-210). Authors should avknowledge that Pearson’s r places most correlations within low correlations category.
Re: Section 3.4 was improved.
7. The degree/strengths of correlations should also be included in the Abstract, for consistency.
Re: The abstract was improved.
8: PMID: 7699656 also has tackled the same topic and problematic. The common aspects and differences to that study should also be included in the Discussion section as well.
Re: tje discussion was improved.
9. 460-480 should report on the degree of correlation (strong, moderate, low), should the case be for each correlation, not only the significance.
Re: Te section was improved.
10. the Conclusion may be rephrased, as correlation (let alone low correlations) do not imply causation. However, the conclusion depicts a causal statement, that is not clear conclusion of the study, as for that another stydy design should be employed (prospective study with diet intervention). This is not the case, therefore the conclusion should be rephrased.
Re: The conclusion was improved.
11. a clearly deliniated Limitations paragraph should be introduced, to state the limitations of the study (small sample, sample sizes criteria, unicentric study, low correlations found, etc).
Re: The limitation was added.
12. also, the authors should clearly state the way they have dealed with multiple comparisons and what adjustments that have employed (e.g. Bonferroni correction) as there are multiple analyais on same sample and without correction the risk of finding a significance where it is actually not a true significance in the data is quite high. All these suggestions are meant to simply improve the manuscript, the quality of the presentation and the data analysis, and I do hope the authors will take them into consideration within an significantly improved version for revision.
Re: 3.4 section was improved, multiple analysis was made.
Round 2
Reviewer 2 Report
Comments and Suggestions for Authors
Most of the comments have been properly addressed. Lines 480-484, it is not sufficient to mention “significant direct correlation”, must be specified the degree (low correlation, moderate correlation, strong correlation).
Grammar and punctuation should be checked in the entire article (e.g. “LiMMitations”)
Grammar and punctuation should be checked in the entire article (e.g. “LiMMitations”)
Author Response
Dear Reviewer,
We wanted to thank you very much for your valuable comments, which improved the quality of the manuscript.
We added the strength of the correlation:
Moreover, authors revealed significant direct moderate correlations between SFA and erythrocyte sedimentation rate (r = 0.445), duration of disease (r = 0.403) and morning stiffness (r = 0.434).
We have corrected the spelling.